# The Glycosylation of Immune Checkpoints and Their Applications in Oncology

**DOI:** 10.3390/ph15121451

**Published:** 2022-11-23

**Authors:** Linlin Zheng, Qi Yang, Feifei Li, Min Zhu, Haochi Yang, Tian Tan, Binghuo Wu, Mingxin Liu, Chuan Xu, Jun Yin, Chenhui Cao

**Affiliations:** 1School of Medicine, University of Electronic Science and Technology of China, Chengdu 610054, China; 2Biotherapy Center, Third Affiliated Hospital of Harbin Medical University, Harbin 150081, China; 3Collaborative Innovation Centre of Regenerative Medicine and Medical BioResource Development and Application Co-constructed by the Province and Ministry, Guangxi Medical University, Nanning 530021, China; 4School of Medical and Life Sciences, Chengdu University of Traditional Chinese Medicine, Chengdu 611137, China; 5School of Basic Medical Sciences, Chengdu University of Traditional Chinese Medicine, Chengdu 611137, China; 6Department of Oncology, Sichuan Academy of Medical Sciences, Sichuan Provincial People’s Hospital, University of Electronic Science and Technology of China, Chengdu 610072, China; 7Sichuan Key Laboratory of Radiation Oncology, Sichuan Cancer Hospital & Institute, Sichuan Cancer Center, School of Medicine, University of Electronic Science and Technology of China, Chengdu 610041, China

**Keywords:** post-translational modifications, glycosylation, immune checkpoints, cancer therapy, GlycoRNA

## Abstract

Tumor therapies have entered the immunotherapy era. Immune checkpoint inhibitors have achieved tremendous success, with some patients achieving long-term tumor control. Tumors, on the other hand, can still accomplish immune evasion, which is aided by immune checkpoints. The majority of immune checkpoints are membrane glycoproteins, and abnormal tumor glycosylation may alter how the immune system perceives tumors, affecting the body’s anti-tumor immunity. Furthermore, RNA can also be glycosylated, and GlycoRNA is important to the immune system. Glycosylation has emerged as a new hallmark of tumors, with glycosylation being considered a potential therapeutic approach. The glycosylation modification of immune checkpoints and the most recent advances in glycosylation-targeted immunotherapy are discussed in this review.

## 1. Introduction

Immune checkpoint molecules, which are expressed on immune cells, have the ability to modulate the amount of immune activation and thus play an essential role in the prevention of the occurrence of autoimmunity. Likewise, tumors are clever enough to evade immune surveillance. Immune checkpoint molecules commonly used include PD-1, CTLA-4, TIM-3, LAG-3, TIGIT, etc. [1,2]. Currently, immunotherapy has attracted much attention. Inhibitors targeting immune checkpoint molecules have shown promising results in the treatment of tumors [3,4], and blocking interactions between immune receptors and ligands has become an effective cancer immunotherapy paradigm.

PTMs (post-translational modifications) refer to the chemical modification of specific amino acid residues in proteins [5]. The biochemical addition of functional groups or proteins occurs following protein biosynthesis, increasing the functional diversity of proteins [6]. PTM, which includes glycosylation, phosphorylation, methylation, acetylation, ubiquitination, and other processes, is critical for influencing protein activity or expression levels [7,8,9,10]. Glycosylation is the most abundant and diverse type of post-translational modification of proteins, and it is an important cellular mechanism for regulating a variety of physiological and pathological functions [11]. Protein glycosylation includes 14 distinct types, including N-Glycosylation, 11 types of O-Glycosylation, C-mannosylation, and generation of glycosylphosphatidylinositol (GPI)-anchored proteins, with N-glycosylation and O-glycosylation being the two most common [12]. Additionally, a groundbreaking recent study discovered that small non-coding RNAs could also be glycosylated [13] (Figure 1).

Glycoproteins can carry out a variety of biological functions correctly. Specific glycoproteins, such as PSA for prostate cancer, CA125 for ovarian cancer, CA19-9 for gastrointestinal and pancreatic cancer, and AFP for liver cancer, can be used as biomarkers in human cancers [14]. Protein glycosylation modifications can help the correct positioning and migration of immune cells. Changes in cell glycosylation not only have an immediate impact on cell growth and survival but also promote tumor-induced immunomodulation and eventual metastasis [15]. Recent studies have revealed that glycosylation is involved in the fundamental molecular and cellular biological processes of tumorigenesis, such as tumor immunomodulation [16]. Tumor cells typically exhibit extensive glycosylation alterations when compared to untransformed tumor cells, which play a crucial role in cancer development and progression [16]. Several studies have shown that glycosylation plays a relatively important role in tumorigenesis and that aberrant glycosylation of proteins is frequently a signal of tumorigenesis. Furthermore, protein glycosylation is also involved in tumor proliferation, invasion, metastasis, and drug resistance [17,18]. In recent years, glycosylation has become a new hallmark of cancer. Researchers are focusing on N-glycosylation [19], and there also have been some new progress in O-glycosylation. Abnormal tumor glycosylation may alter how the immune system perceives tumors, thereby affecting the body’s anti-tumor immunity, and targeting glycosylation has emerged as a potential therapeutic approach [20].

This review focused on the glycosylation modification of immune receptors/ligands involved in tumor immunity, as well as the most recent advances in glycosylation-targeted immunotherapy.

## 2. The Process of Glycosylation of Proteins

Glycosylation is produced by complicated biosynthetic pathways that include hundreds of glycosyltransferases, glycosidases, transcription factors, transporters, and protein backbones [21]. The main process involves the transfer of the sugar chain to the protein via glycosyltransferase catalysis and the formation of glycoside bonds between glycan and amino acid residues. The assembly of glycosylated proteins is completed through a series of transport, sugar chain end shearing, and fucosylation or sialylation [22]. Most glycosylation occurs in the ER (endoplasmic reticulum) and ends in the Golgi apparatus. However, there are very few that start from the Golgi apparatus, with the exception of mucin-type O-glycosylation [23].

As the junction points, N-Glycosylation forms an N-glycosidic bond with the amide group of asparagine, the α-amino group of the N-terminal amino acid, and the ω-amino group of lysine or arginine. The multiunit oligosaccharyltransferase (OST) complex, which contains two catalytic subunits, SST3A and STT3B, catalyzes this process [24]. It is the most abundant and well-conserved protein modification in the ER of eukaryotic cells and plays an important role in protein folding, assembly, and transport [25,26]. In order to maintain the stability and quality of proteins, misfolded proteins in the ER are transferred to the cytoplasmic matrix for proteasomal degradation via ER-associated degradation (ERAD). Misfolded proteins that are resistant to ERAD are too large to cross the ER membrane and are delivered to the endo-lysosome for ER-to-lysosome-associated degradation (ERLAD) [27,28]. Additionally, both degradation pathways have been shown to be associated with N-glycans [29].

O-glycosylation occurs in the Golgi apparatus and forms O-glycosidic bonds as junction points with the hydroxyl groups of serine, threonine, hydroxylysine, and hydroxyproline. O-glycosylation is more direct and heterogeneous than N-glycosylation. O-GlcNAcylation, also known as mucin-type O-glycosylation, is the most common type of O-Glycosylation. Other less common types include O-Fucosylation, O-GluNAcylation, and O-Mannosylation [16]. Many proteins are modified by reversible O-glycosylation, such as transcription factors, cytoskeletal proteins, oncogenes, and kinases [30]. O-GlcNAc modification is achieved by uridine diphosphate N-acetylglucosamine (UDP-GlcNAc), which is the end product of the hexosamine biosynthetic pathway (HBP) [31]. The activity of O-GlcNAc transferase (OGT) mediates the O-GlcNAcylation of proteins. The enzyme that removes glycans is β-N-acetamidoglucosidase (O-GlcNAcase, OGA) [32]. Crosstalk with other shared PTM codes, particularly phosphorylation on the same Ser/Thr residues, affects protein complex formation and signal transmission in part [33]. It can be found on serine and threonine residues, as well as tyrosine, hydroxylysine, and hydroxyproline. O-glycosylated mucins are expressed in a range of tumor cell types and induce various oncogenic signaling molecules to promote tumor progression by modulating protein stability [34,35].

## 3. N-Glycosylated Immune Checkpoints

Most immune checkpoint molecules can be modified by glycosylation, with the B7-CD28 superfamily being particularly notable. It belongs to the co-stimulatory molecule families and plays a vital role in tumor immunotherapy. These co-stimulatory pathways can either provide agonistic signals to enhance and maintain T-cell immune responses or provide inhibitory signals to suppress T-cell immunity responses [36]. There are also some additional molecules that can modulate tumor immunity mediated by glycosylation. In Figure 2, we summarized the immune receptors and ligands regulated by glycosylation on the surface of immune and tumor cells.

### 3.1. B7-CD28 Superfamily

The “two-signal model”, proposed in 1970, is required for T cell proliferation and activation: the first signal is initiated by the T-cell-receptor (TCR) recognizing the antigen peptide-major histocompatibility complex (MHC) on the surface of antigen-presenting cells (APCs); the second signal is a co-stimulatory signal that is mediated by the interaction of co-stimulatory molecules between T cells and APCs [37].

The earliest and most significant sets of co-stimulatory molecules in the B7-CD28 superfamily are CD28 and CD80/CD86. CD28 is a glycoprotein found on the surface of T lymphocytes that binds to CD80 (B7-1) and CD86 (B7-2). When N-glycosylation was prevented by a point mutation in the N-Glycosylation site of CD28 or reduced by glycosidase inhibitors, the binding of CD28-CD80 was increased significantly, and downstream signaling activation was amplified, indicating that N-linked glycosylation negatively regulates CD28 function [38]. Similarly, CTLA-4 (cytotoxic T-lymphocyte-associated protein 4), commonly known as CD152, is a homolog of CD28, but it binds to CD80/CD86 (B7-1/2) with a higher affinity than CD28. CTLA-4 is an immune checkpoint receptor expressed on T cells and competes with CD28 for binding to CD80/CD86 on APCs, resulting in the downregulation of immune responses [39,40]. Moreover, CTLA-4 can also be regulated by glycosylation. The N-glycan number and branching of the glycoprotein can influence its cell-surface levels. TCR signaling upregulates CTLA-4 surface expression to affect T-cell function via hexosamine and N-glycan branching pathways [41].

At present, the programmed cell death protein-1 (PD-1)/programmed death-ligand 1 (PD-L1) is still one of the major immune checkpoints identified [42]. Tumor immune escape mediated by the PD-1/PD-L1 signal pathway is an essential mechanism of tumorigenesis and progression [43]. PD-1(B7-H1) is an inhibitory receptor expressed on the surface of activated T cells that inhibits T cell receptor (TCR)/CD28 signaling by attaching to its cancer-cell ligand PD-L1 [44]. In the tumor microenvironment, tumor cells express PD-L1, which interacts with PD-1 on T cells to influence the anti-tumor activity of effector T cells. Inhibition or blockade of the PD-1/PD-L1 pathway significantly enhances T cell-mediated immune killing activity and, consequently, anti-tumor effect. PD-1/PD-L1-mediated immunosuppression is strongly correlated with its own protein expression levels and post-translational modifications.

Glycosylation is essential for the interaction of PD-1 and PD-L1. By using the genome-wide loss-of-function screening strategy based on the CRISPR-Cas9 system, researchers identified genes involved in the core fucosylation pathway as positive regulators of cell-surface PD-1 expression. N49 and N74, of the core fucosylated N-glycans for PD-1, are determined as the significant locations, inhibition of Fucosyltransferase 8 (Fut8), a key fucosyltransferase, reduced PD-1 expression and enhanced T cell activation [45]. A recent study also indicated that PD-1 is highly N-glycosylated in T cells at N49, N58, N74, and N116 sites, especially at N58, which is essential for PD-L1/PD-L1 binding. When T cells are activated, glycoforms of PD-1, such as poly-LacNAc and core fucose, are upregulated. They verified that β-1,3-N-acetylglucosaminyl-transferase (B3GNT3) and FUT8 catalyze the synthesis of poly-LacNAc and core fucose, respectively [46], which is consistent with the aforementioned experimental findings [45]. Glycosylation can also regulates PD-L1 expression in cancer stem cells. Cancer stem cells (CSCs) are a subset of tumor cells that possess self-renewal ability and are implicated in tumorigenesis, progression, drug resistance and metastasis [47]. N-glycosyltransferase STT3, which is upregulated by epithelial–mesenchymal transition (EMT) through β-catenin, induces PD-L1 N-glycosylation to regulate PD-L1 accumulation in CSCs, hence encouraging immune evasion [48]. High expression of β-catenin and STT3A/B has been linked to a poor prognosis in colon cancer patients. KYA1797K, a tumor suppressor gene in colon cancer, inhibits PD-L1 N-glycosylation and suppresses immune evasion of colon CSCs through inhibition of the β-catenin/STT3 signaling pathway, opening up new possibilities for colon cancer immunotherapy [49].

Additionally, N-glycosylation plays a critical role in maintaining the stability of proteins. Incompletely glycosylated PD-1 is also regulated by ubiquitination. KLHL22 is an E3 ligase adaptor of PD-1 to induce incompletely glycosylated PD-1 degradation before its membrane localization. Knockdown of KLHL22 inhibits T-cells activation and anti-tumor response [50]. Non-glycosylated PD-L1 is also less stable, and it can bind to glycogen synthase kinase 3β (GSK3β) and be degraded by the ubiquitin/proteasome system after phosphorylation. Glycosylated PD-L1 induced by the EGF signal can hinder the interaction between GSK3β and PD-L1 from stabilizing PD-L1 and enhancing its immunosuppressive activity. An EGFR small-molecule inhibitor, Gefitinib, can inhibit EGF signaling to destabilize PD-L1 and boost anti-tumor immunity [51]. Moreover, the follow-up study demonstrated that EGF signaling also encourages PD-L1 N-glycosylation at N192 and N200 sites in triple-negative breast cancer (TNBC) cells by upregulating B3GNT3, which enables PD-L1 to interact with PD-1 on cytotoxic T cells and causes T cells exhaustion [52]. Additionally, it is reported that metformin has tumor suppressor function [53]. Adenosine 5′-monophosphate (AMP)-activated protein kinase (AMPK) located in the lumen of ER is activated by metformin and phosphorylates PD-L1 at the S195 site. Phosphorylation of PD-L1 induces its aberrant N-glycosylation and downregulation via ER accumulation and ERAD, leading to enhance CTL activity against tumor cells [54]. Similarly, in prostate and triple-negative breast cancer cells, the Sigma1 activator could upregulate PD-L1 expression by promoting the glycosylation of PD-L1, and the Sigma1 inhibitor could sequester and eliminate PD-L1 by autophagy, thus preventing functional PD-L1 expression at the cell surface [55].

A growing amount of clinical data suggests that PD-L1 expression is an important biomarker for predicting the efficacy of PD-1 inhibitors. Currently, the standard clinical method for detecting PD-L1 is an immunohistochemical (IHC) examination. In general, the higher the expression of PD-L1, the better the efficacy of PD-1 inhibitors is likely to be. However, a proportion of patients with low or no PD-L1 expression can respond to PD-L1 inhibitors as well. This may be related to the glycosylation status of PD-L1. Though PD-L1 is highly glycosylated, diagnostic antibodies used to detect PD-L1 clinically do not effectively recognize glycosylated PD-L1, resulting in false negatives in PD-L1 IHC examinations. Removing the glycans of PD-L1 from Formalin-fixed paraffin-embedding (FFPE) blocks of tumors with peptide-N-glycosidase F (PNGase F) followed by IHC examination significantly improved the sensitivity of antibody-based detection. Deglycosylated PD-L1 is more beneficial for guiding clinical treatment and predicting clinical outcomes of patients [56].

Apart from PD-L1, PD-L2, also known as CD273 or B7-DC, is the second important ligand that binds to PD-1. The binding of PD-L2 and PD-1in cancer plays a role in inhibiting the activation of T cells [57]. The expression level of PD-L2 can be considered supplementary information to predict the clinical efficacy of anti-PD-1 therapy together with the PD-L1 expression level [58,59]. It was shown that PD-L2 is N-glycosylated mediated by STAT3-FUT8 at N64, N157, N163, and N189 sites and overexpressed in Cetuximab-resistant head and neck squamous cell carcinoma (HNSCC). Moreover, N-glycosylation is crucial for maintaining PD-L2 stability via blocking ubiquitination-mediated lysosomal degradation. Inhibition of PD-L2 glycosylation reduces epidermal growth factor receptor (EGFR) blocker cetuximab resistance in an in vivo model of HNSCC, and glycosylation levels can predict anti-EGFR efficacy [60]. The mechanisms of PD-1 and PD-L1/2 binding and degradation associated with glycosylation are shown in Figure 3.

As mentioned above, co-stimulatory signals are necessary for T-cell activation. The inducible co-stimulator (ICOS) binds to ICOSL (B7-H2) and provides a co-stimulatory signal to activate T cells. ICOS/ICOSL is a pair of members of the CD28/B7 superfamily [61,62]. The presence of three speculative N-linked glycosylation sites in ICOS has been demonstrated, and N89-linked glycosylation keeps ICOS stable and delivers it to the cell surface membrane [63]. The structure of ICOS/ICOSL was recently analyzed, and it was discovered that the N110 glycosylation site in ICOS is crucial in ICOS-L binding. Knockdown of the N110 site resulted in a 4.3-fold increase in binding affinity to ICOSL [64].

The B7-CD28 immune checkpoint family can be phylogenetically divided into three groups. The third group includes recently identified B7-H3 (CD276), B7-H4 (B7x/B7S1), HHLA2 and TMIGD2 [65]. B7-H3 is expressed in a variety of tumors and is associated with poor prognosis [66]. For example, inhibition of B7-H3, which is markedly overexpressed in tumor cells and APCs in oral squamous cell carcinoma, can prevent the formation of the tumor. Furthermore, when compared with normal oral epithelial cells, B7-H3 on the surface of tumor cells is highly fucosylated and aberrantly N-glycosylated, and it is likely that the glycosylation of B7-H3 affects tumor biological characteristics [67]. According to a recent study, FUT8 catalyzes core fucosylation at N-glycans of B7-H3 to maintain its high expression and stability, thereby promoting immunosuppression in TNBC patients [68]. Similarly, B7-H4, a co-inhibitory ligand, is highly expressed in immune-cold tumors and negatively correlates with PD-L1 expression. A recently published study found that N-linked glycosylation of B7-H4 is essential to preserve stability by preventing B7-H4’s ubiquitination. Inhibiting N-glycosylation of B7-H4 turns immune-cold tumors into immune-hot tumors, increasing response to immunotherapy [69]. 

Unlike T cells, natural killer cells (NK cells) are a type of lymphocyte that has the capacity to destroy tumor cells without prior sensitization. Mature NK cells constitutively express multiple activating receptors, the most important of which are natural cytotoxicity receptors (NCRs), including NKp30, NKp44, and NKp46. NKp30 is a primary activating receptor of NK cells that binds to B7-H6. It has been reported that the stalk domain of NKp30 is essential for ligand binding that can be regulated by glycosylation. Differential glycosylation of NKp30 affects the ligand-binding affinity and thus affects downstream intracellular signaling. In addition, three different sites (N42, N69, and N121) are N-glycosylated in the ectodomain of NKp30 [70]. Recently, research has reported that N-glycosylation is crucial to the oligomerization of NKp30 [71].

### 3.2. Immune Checkpoints Outside the B7 Superfamily

In addition to the CD28/B7 superfamily, several membrane receptors are involved in T-cell immune responses, many of which are regulated by N-glycosylation. Various co-inhibitory receptors expressed by immune cells, tumor cells, and immunosuppressive bone marrow cells can also be controlled by N-glycosylation.

Lymphocyte activation gene-3 (LAG-3) is an important co-inhibitory receptor. LAG-3 is a type I transmembrane protein expressed in T cells with four structural domains. The extracellular structural domain of LAG-3 is similar to that of CD4, and it can bind to MHCII competitively with CD4 to inhibit T-cell activation. LAG-3 is regarded as a prospective therapeutic target for tumor immunotherapy since several studies have demonstrated a correlation between high levels of LAG-3 expression with tumor growth and prognosis in a variety of human tumor types [72,73]. In addition, LAG-3 is highly glycosylated as there are multiple N-glycosylation sites in D2-D4 domains. It was reported that LAG-3 could interact with Galectin-3 or liver sinusoidal endothelial cell lectin (LSECtin) and that their binding sites are glycans on LAG-3. However, it is less clear whether LAG-3 glycosylation is associated with tumor immunity [74,75].

T cell immunoreceptor with immunoglobulin and ITIM domains (TIGIT) is another crucial immune inhibitory receptor expressed on T cells and NK cells. TIGIT contains three ligands, of which PVR (CD155), expressed in tumor cells and APC, is the main ligand for TIGIT with high affinity. TIGIT ultimately suppresses anti-tumor immunity through binding to PVR and a subsequent series of responses [76,77,78]. Researchers have been striving to explore the potential regulatory mechanisms of PVR/TGIT interactions. TIGIT is N-linked glycosylated and PNGase F treatment to remove N-glycans from TIGIT inhibits TIGIT/PVR binding. Two N-glycosylation sites, N32 and N101, were discovered in more detail, with N101 having a more significant impact on the interaction between PVR and TIGIT. The above findings suggest that N-glycosylation of TIGIT is essential for maintaining the interaction between TIGIT and PVR, which may be a potential target for immunotherapy [79].

4-1BB (CD137/TNFRSF9) is an inducible tumor necrosis factor receptor (TNFR) superfamily member expressed by various cells of the immune system, particularly by T cells [80]. The crystal structure reveals that 4-1BB interacts with its ligand, 4-1BBL/CD137L/TNFSF9, through the cysteine-rich domains (CRD) 2 and 3, promoting the activation and proliferation of immune cells [81,82]. However, a more recent study identified that the extracellular domain of 4-1BB has two glycosylation sites (N138 and N149) located on CRD4, confirming that N-glycosylation is not associated with 4-1BB/4-1BBL binding. Furthermore, N-Glycosylation facilitates 4-1BB membrane localization by maintaining its oligomerization [83].

T-cell immunoglobulin and mucin domain-3 (TIM-3) is an activation-induced inhibitory receptor and binds to its ligand galectin-9 (Gal-9) to induce immune tolerance and T-cell exhaustion in cancer [84]. Simultaneously, the co-expression of PD-1 and TIM-3 on T cells has a bearing on T cell exhaustion in cancer [85]. It was reported that Gal-9 binds carbohydrate chains on TIM-3, and TIM-3 treated with recombinant glycosidase (PNGase F) fails to bind Gal-9, suggesting it is glycosylation that regulates the binding [84]. Interestingly, Co-expressed PD-1 coupled with TIM-3 to form heterodimers, and Gal-9 crosslinks these dimers to form galectin/glycoprotein lattices, which were then able to block TIM-3/Gal-9-induced T cell death [86]. 

The sugar chains of glycoconjugates are often terminated by multiple sialic acids, which play a key role in cell interactions, tumor cell dissociation, and metastasis. Sialic acid-binding immunoglobulin-like lectin (Siglec) family functions as an inhibitory receptor to shape the immunosuppressive tumor microenvironment by recognizing sialic acid glycans, albeit the precise ligand is still unknown [87]. Siglec-15, different from other Siglec family members [88], shares more than 30% homology with B7-H1 and other B7 family members, although Siglec-15 expression is independent of that of B7-H1 expression. Siglec-15 is upregulated in human cancer cells and tumor-associated macrophages/myeloid cells [89]. Siglec-15 was identified to be N-linked glycosylated on the N172 site, leading to decreasing its lysosome-dependent degradation and promoting its cell membrane localization [90]. In addition, tumor cells can express the “don’t eat me” signal protein such as CD47 [91] and CD24 [92] on the cell membrane to avoid the phagocytosis of macrophages and achieve immune escape. CD47, also known as integrin-associated protein (IAP) or Antigenic surface determinant protein OA3, is a glycoprotein with an immunoglobulin-like structure that is widely expressed in tumor cells. It binds to signal regulatory protein α (SIRPα) on the surface of macrophages, resulting in a series of cascade reactions that inhibit macrophage phagocytosis [93,94]. Six possible N-glycosylation sites on CD47 were identified in certain studies, and removal of the sites alters CD47 expression. However, whether N-glycosylation affects CD47/ SIRPα binding and tumor immune evasion remains unknown [95,96]. A brand-new “don’t eat me” signal protein, CD24, is similar to CD47. It is highly glycosylated and expressed on the surface of tumor cells and acts with the inhibitory receptor Siglec-10, which is highly expressed on tumor-associated macrophages (TAMs), thereby inhibiting the phagocytosis of macrophages [92]. A 3D model of the Siglec-10/sialoglycans complex was built and unveiled that Siglec-10 favored sialylated complex-type N-glycans on substrates for binding [97].

Another activating receptor for NK cells is the natural killer group 2D (NKG2D), and it forms a homodimer expressed on the surface of NK cells. Due to cellular stress and DNA damage, MHC class I polypeptide-related sequence A (MICA) and MHC class I polypeptide-related sequence B (MICB) are expressed on the tumor cell surface, which can bind to NKG2D to induce an immune response [98]. However, tumors evade immune surveillance through the proteolytic shedding of MICA proteins [99]. For cell membrane surface expression, N-glycosylation of proteins on the extracellular domain expressed by certain MICA alleles is essential [100,101].

2B4 (CD244), a member of the signaling lymphocyte activation molecule (SLAM)-related receptor family, is a cell surface glycoprotein that is expressed on all NK cells, CD8+ T cells, and dendritic cells (DC) and involved in the regulation of NK cells and T cells function [102]. CD48 is expressed on the surface of hematopoietic cells to act as a ligand for 2B4, and CD48/CD244 binding is absolutely vital for cell-cell interactions [103]. 2B4 is reported to be modified by N- and O-glycosylation and highly sialylated. The N-glycosylation of 2B4 is important for its interaction with CD48, and the removal of N-glycan by glycosidase will markedly inhibit the binding of 2B4 and CD48 [104]. Moreover, the removal of sialic acid from the NK cell’s surface or inhibition of O-linked glycosylation increased 2B4/CD48 binding and thus promoted NK cell activation [104].

## 4. O-Glycosylation

It was reported that O-Glycosylation encourages tumor growth and development during tumor progression. TIM-3 is mucin-type O-glycosylated, which is necessary for Gal-9 binding. However, whether it plays a role in the diagnosis and treatment of tumors is unclear [105]. Serine and threonine are indispensable links in the process of glycosylation. Post-translational modification of serine/arginine-rich protein kinase 2 (SRPK2) by O-Glycosylated promotes the growth of breast cancer cells in vitro and in vivo [106]. Additionally, it has been certified that O-glycosylated phosphoglycerate kinase 1 (PGK1) enhances glycolysis pathways. Glycosylation modification inhibits the intra-mitochondrial tricarboxylic acid (TCA) cycle by inducing mitochondrial translocation of PGK1, thereby promoting the proliferation and growth of colon cancer [107].

## 5. GlycoRNA

Regarding glycosylation, the default is often the glycosylation of proteins or lipids. However, recent studies demonstrated that RNA, in addition to proteins and lipids, may be employed as a third glycosylation target. The newly identified RNA that can be glycosylated is a small, conserved non-coding RNA called “glycoRNA”, which is typically located in the cytoplasm. Small non-coding RNAs include snRNA, snoRNA, miRNA, siRNA, piRNA, stRNA, etc. GlycoRNA features N-glycans that are rich in sialic acid and fucose [13]. The specific pathway of glycoRNA chemical synthesis is unclear, but it is likely to be in much the same way as protein N-glycosylation, even requiring some of the same enzymes [108]. GlycoRNA appears on the extracellular membrane, suggesting it may play a role in signal transduction. Due to the richness of sialic acid, GlycoRNA binds to two sialic acid-binding immunoglobulin-like lectins (Siglecs) [109]. GlycoRNA is proposed to act as a potential ligand for the Siglec family [13]. GlycoRNA has been detected in various human cell lines, and these RNAs on specific cells have been found to be identical to small RNAs associated with a variety of autoimmune diseases, making it highly likely that GlycoRNA is relevant to the immune system [13]. This significant finding may bring us a new way to think about the relationship between tumors and immune checkpoint molecules to find new ways for tumor immunotherapy.

## 6. Applications in Glycosylation-Targeted Immunotherapy

Immunotherapy has made enormous strides in recent years. Immune checkpoint inhibitory antibodies against PD-1/PD-L1, including Pembrolizumab, Nivolumab, and Atezolizumab, have shown significant clinical benefit, although overall remission rates are still modest. More targets and drugs need to be explored and developed. Most immune checkpoint molecules can be modified by glycosylation. Therefore, glycosylation of immune checkpoint molecules has become a promising target for tumor immunotherapy. New drugs targeting glycosylation have been developed with in vivo and in vitro trials underway, and some drugs have already been initiated in clinical trials. In Table 1, we summarize the novel drugs targeting glycosylation. 

PD-1 and PD-L1 have multiple N-glycosylation sites, and glycosylation modifications significantly affect the efficacy of immunotherapy. Based on this, STM418, a monoclonal antibody targeting the N58 glycosylation site of PD-1 was developed, exhibited a higher binding affinity for PD-1 than FDA-approved nivolumab and pembrolizumab, effectively inhibiting PD-1/PD-L1 binding and enhancing anti-tumor immunity [46]. Researchers obtained a specific monoclonal antibody MW11-h317 against PD-1 by humanizing the antibody after immunizing mice with recombinant human PD-1. Analysis of the binding site revealed that it mainly targeted the N58 glycosylation site of PD-1. MW11-h317 effectively blocks the binding between PD-1 and PD-L1/L2, inducing T-cell-mediated immune response and inhibiting tumor growth in vivo trials [110]. MAb059c, a monoclonal antibody against PD-1, also recognizes the N58 glycosylation site of PD-1. In addition, anti-PD-1 nivolumab and pembrolizumab were found to be independent of N-glycosylation [111]. Cemiplimab was the first to be approved for locally advanced or metastatic cutaneous squamous cell carcinoma in 2018. It is found that the N58-glycan of PD-1 may be essential for cemiplimab to exert its anti-tumor effect through analyzing the crystal structure of PD-1 and cemiplimab. When the N58-glycan was removed, the binding affinity of cemiplimab with PD-1 was significantly reduced, and the efficacy was weakened [112]. Currently, a number of clinical trials on cemiplimab in other tumor types are underway. Similarly, the glycosylation site of PD-L1 can also be used as a therapeutic target. STM108, a monoclonal antibody, was isolated and purified to target B3GNT3-mediated glycans on N192 and N200 glycosylation sites of PD-L1, inducing PD-L1 internalization and degradation [52]. Except for monoclonal antibodies, the small molecule PD-L1 inhibitor BMS1166 specifically inhibits PD-L1 glycosylation, preventing PD-L1 from being transported from the endoplasmic reticulum to the Golgi apparatus. Hypoglycosylated PD-L1 accumulates within the ER and enters the ERAD pathway for degradation, thereby blocking PD-L1/PD-1 signaling and activating T cells [113].

Targeting enzymes in the glycosylation process can also have a therapeutic effect. As mentioned before, PD-1 is highly N-glycosylated in T cells, and Most of the N-glycans on PD-1 are fucosylated, especially in exhausted T cells. The core fucosylation inhibitor 2-fluoro-L-Fucose (2F-Fuc) can inhibit Fut8, thereby reducing PD-1 expression and enhancing T-cell activation [45,46]. Furthermore, the combination of 2F-Fuc and anti-PD-L1 improves the efficacy of TNBC tumors with high B7-H3 expression [68]. NGI-1 is a small molecule inhibitor of OST that inhibits the activity of STT3A and STT3B. Compared to STT3A, NGI-1 shows a higher specificity for STT3B [114,115]. Inhibition of STT3A with NGI-1 inhibits the proliferation, invasion, and migration of lung adenocarcinoma cells [116]. NGI-1 also inhibits B7-H4 glycosylation, leading to its ubiquitinated degradation, and in combination with PD-L1 inhibitors, it can inhibit the growth of immune-cold tumors such as TNBC [69]. Glucose analog 2-deoxy-glucose (2-DG), a competitive inhibitor of hexokinase to act as a glycolytic inhibitor, is able to interfere with the N-glycosylation synthesis of proteins. 2-DG inhibits N-linked glycosylation of MICA to prevent its cell surface expression [100]. Although Poly ADP-ribose polymerase (PARP) inhibitors have shown initial success in tumors, they promote PD-L1 expression to induce immune escape. 2-DG can trigger PD-L1 deglycosylation, thus reversing the side effects of PARP inhibitors [117]. In addition, 2-DG was combined with the EGFR inhibitor gefitinib to inhibit PD-L1 expression and PD-1/PD-L1 binding from enhancing anti-tumor immunity [118].

Therapeutic antibodies targeting immune checkpoints can also be modified by glycosylation. Antibodies consist of fragment antigen binding (Fab), which binds to a specific antigen and thus determines the specificity and affinity of the antibody, and fragment crystallizable (Fc), which binds to the Fc receptor expressed on the surface of immune cells and thus activates the immunological effect. Antibody-dependent cell-mediated cytotoxicity (ADCC) is initiated to kill tumor cells when antibodies bind with both the antigen present on the tumor cell and Fc receptors on the effector cells. There are three main types of Fc receptors: FcγRI, FcγRII, and FcγRIII. FcγRIII can be subdivided into FcγRIIIa, and FcγRIIIb, of which FcγRIIIa is usually considered to be the key receptor. The N-glycosylation modification of antibodies consists of N-GlcNAc, mannose, fucose, and galactose, of which fucose is considered to be the most important type affecting ADCC activity. Defucosylation significantly increases the affinity of antibodies/FcγRIIIa and ADCC activity [119,120]. SEA-TGT is an anti-TIGIT non-fucosylated monoclonal antibody based on proprietary sugar-engineered antibody (SEA) technology. It enhances FcγRIIIa affinity and ADCC activity. The combination of SEA-TGT and anti-PD-1 depletes T cells with high expression of TIGIT and reverses T cells exhaustion, thereby enhancing anti-tumor immunity. In order to test the safety and efficacy of SEA-TGT in advanced cancer, a phase 1 clinical trial (NCT04254107) is recruiting subjects [121].

**Table 1 pharmaceuticals-15-01451-t001:** Novel drugs in glycosylation-targeted immunotherapy.

Reference	Drug Name	Drug Type	Target	Study Phase
[44]	STM408	Monoclonal antibody	N58-glycosylated PD-1	Preclinical study
[108]	MW11-h317	Monoclonal antibody	N58-glycosylated PD-1	Preclinical study
[109]	mAb059c	Monoclonal antibody	N58-glycosylated PD-1	Preclinical study
[50]	STM108	Monoclonal antibody	N-glycosylated PD-L1	Preclinical study
[111]	BMS1166	Small molecule	N-glycosylated PD-L1	Preclinical study
[43,44,66]	2F-Fuc	Glucose analog	FUT8	Preclinical study
[67,112,113,114]	NGI-1	Small molecule	STT3A/B	Preclinical study
[98,115,116]	2-DG	Glucose analog	Hexokinase	Preclinical study
[119]	SEA-TGT	Monoclonal antibody	TIGIT	Clinical trial

## 7. Conclusions and Perspectives

Proteins are the building blocks of life, and different types of glycosylation broaden the functional diversity of proteins. Protein glycosylation is involved in tumorigenesis, proliferation, invasion, metastasis, and drug resistance. Aberrant glycosylation of immune checkpoint molecules enables tumors to evade the body’s immune surveillance to achieve immune evasion, revealing the importance of glycosylation. A growing number of studies have focused on the relationship between glycoproteins, and the immune system, most of which are N-glycoproteins since O-glycosylation without a common core structure is more complex than N-glycosylation. Surprisingly, we found that RNA could be modified by glycosylation, which made RNA the third scaffold for glycosylation except for proteins and lipids. More molecules and structures that can be modified by glycosylation may be discovered in the future.

Exploring the molecular mechanisms of glycosylation will deepen our understanding of tumor immunity. It will also be of great value to further investigate the crosstalk between glycosylation and other post-translational modifications. Rapid advances in glycomics will allow for the development of immunotherapeutic drugs based on inhibitors or glycan antagonists. An in-depth study of glycosylation of immune checkpoint molecules may provide a better strategy for the clinical application of tumor immunotherapy.

## Figures and Tables

**Figure 1 pharmaceuticals-15-01451-f001:**
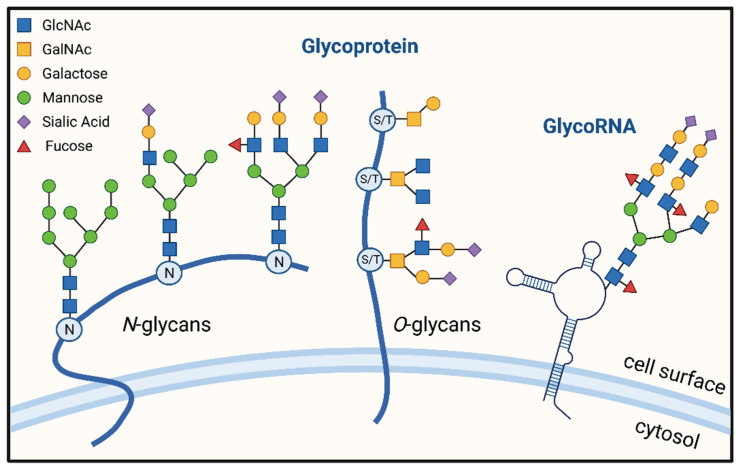
Model diagram of protein glycosylation (N-glycosylation and O-glycosylation) and RNA glycosylation. Created with BioRender.com on 14 October 2022.

**Figure 2 pharmaceuticals-15-01451-f002:**
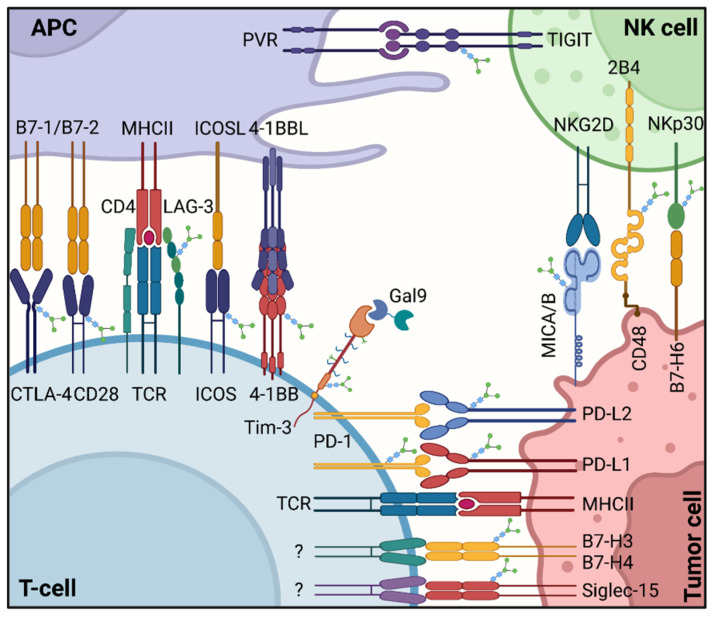
The immune receptors and ligands regulated by glycosylation on the surface of immune and tumor cells. Depicted are various ligand–receptor interactions between immune cells and tumor cells. One important family of membrane-bound ligands that bind both co-stimulatory and inhibitory receptors is the B7 family. All of the B7 family members and their known ligands belong to the immunoglobulin superfamily. Created with BioRender.com on 14 October 2022.

**Figure 3 pharmaceuticals-15-01451-f003:**
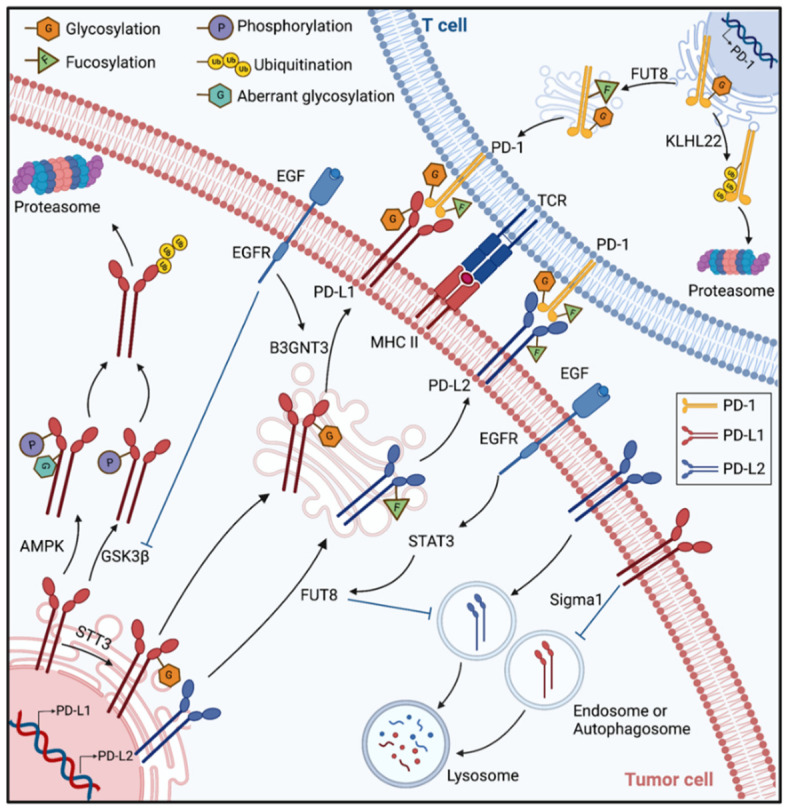
The mechanisms of PD-1 and PD-L1/2 binding and degradation associated with glycosylation. Created with BioRender.com on 14 October 2022.

## Data Availability

Not applicable.

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
