# Peer review of "The Glycosylation of Immune Checkpoints and Their Applications in Oncology"

_pharmaceuticals, 2022, doi:10.3390/ph15121451_

Round 1

Reviewer 1 Report

In this manuscript the authors made an exhaustive revision of various forms of glycosylation which induces modification of the functionality of proteins related to tumor immunity thus affecting immune receptors and ligands involved in cell proliferation, invasion, metastasis, and drug resistance of tumor cells. The study also refers to the unexpected role of glycosylated RNA in the tumorigenic process. The review points out the importance of aberrant glycosylation of immune checkpoint molecules, which enable tumors to achieve immune evasion. Finally the authors mention the expectation of possible advances in glycosylation targeted immunotherapy.

Some considerations and aspects to be clarified by the authors:

1. The work brings important information concerning glycosylation of immune molecules involved in tumor immunity. Except for the broken statements in the introduction chapter, and for some phrases showing finalism expression such as …”tumors use…” (line 35) or  …”mammals can use…” (line 328),  the manuscript is well written and thus the review deserves publication. 

2.   The review greatly covers the literature but surprisingly there are no original contributions by the authors. Most importantly, some sentences may induces the reader to believe it is their contribution, as for instance …”previous studies…”(lines 286-288), or …”as described before…” (lines 316 - 318). Is there any one among the authors working in this interesting research subject? I would expect this point to be clarified.

3.   The manuscript is full of abbreviations and thus the creation of an abbreviation session is recommended;

4.    Points to be corrected:

i)              line 23: However and not however;

ii)             line 111: Item 3 should be in bold;

iii)           lines 154-155: reference Sun et al should be  Sun et al, 2020 (40); the expression …”defined that” should be …”indicated that…”;

iv)           line 171: …”in-mune…”;

v)            line 187: there are two references of Li et al (2016 and 2018) which one is referred to? It should be corrected In the same way as indicated above in item 3.iii;

vi)           line 214: Reference Edurne Rujas et al, should be written as indicated above;

vii)         line 270: “The sugar chains of sugar conjugates…” couldn’t be change to “The sugar chains of glycoconjugates?…”

Reviewer 2 Report

The review article is good. only thing is similar review article is already published in another MDPI journal

https://www.mdpi.com/2073-4409/10/5/1100

Glycosylation of Immune Receptors in Cancer Ruoxuan Sun, Alyssa Min Jung Kim and Seung-Oe Lim *

Authors need to clarify this. 

Round 2

Reviewer 2 Report

Authors have substantially answered the comments.